# An Integrative miRNA-mRNA Expression Analysis Reveals Striking Transcriptomic Similarities between Severe Equine Asthma and Specific Asthma Endotypes in Humans

**DOI:** 10.3390/genes11101143

**Published:** 2020-09-28

**Authors:** Matthias F. Hulliger, Alicja Pacholewska, Amandine Vargas, Jean-Pierre Lavoie, Tosso Leeb, Vincent Gerber, Vidhya Jagannathan

**Affiliations:** 1Swiss Institute of Equine Medicine, Department of Clinical Veterinary Medicine, Vetsuisse Faculty, University of Bern, 3012 Bern, Switzerland; matthias.kraft@vetsuisse.unibe.ch (M.F.H.); alicja.pacholewska@uk-koeln.de (A.P.); vinzenz.gerber@vetsuisse.unibe.ch (V.G.); 2Institute of Genetics, Vetsuisse Faculty, University of Bern, 3012 Bern, Switzerland; tosso.leeb@vetsuisse.unibe.ch; 3Faculty of Veterinary Medicine, University of Montreal, 3200 rue Sicotte, St-Hyacinthe, QC J2S 2M2, Canada; amandine.vargas@umontreal.ca (A.V.); jean-pierre.lavoie@umontreal.ca (J.-P.L.)

**Keywords:** RNA-seq, regulatory networks, differential expression analysis, *Equus caballus*, lung tissue, immunology, precision medicine

## Abstract

Severe equine asthma is an incurable obstructive respiratory condition affecting 10–15% of horses in temperate climates. Upon exposure to airborne antigens from hay feeding, affected horses show neutrophilic airway inflammation and bronchoconstriction, leading to increased respiratory effort. The resulting implications range from welfare concerns to economic impacts on equestrian sports and horse breeding. Immunological and pathophysiological characteristics of severe equine asthma show important parallels with allergic and severe neutrophilic human asthma. Our study aimed at investigating regulatory networks underlying the pathophysiology of the disease by profiling miRNA and mRNA expression in lung tissue samples from asthmatic horses compared with healthy controls. We sequenced small RNAs and mRNAs from lungs of seven asthmatic horses in exacerbation, five affected horses in remission, and eight healthy control horses. Our comprehensive differential expression analyses, combined with the miRNA–mRNA negative correlation approach, revealed a strong similarity on the transcriptomic level between severe equine asthma and severe neutrophilic asthma in humans, potentially through affecting Th17 cell differentiation. This study also showed that several dysregulated miRNAs and mRNAs are involved in airway remodeling. These results present a starting point for a better transcriptomic understanding of severe equine asthma and its similarities to asthma in humans.

## 1. Introduction

Severe equine asthma (sEA) is an incurable obstructive respiratory condition affecting approximately 10% to 15% of horses in temperate climates [1]. When exposed to airborne antigens, mainly during hay feeding, affected horses develop the characteristic signs of sEA exacerbation: neutrophilic airway inflammation, bronchial hyperreactivity, bronchoconstriction, and mucus hypersecretion. These chronic or recurrent exacerbations manifest with coughing, exercise intolerance, and increased respiratory effort. Marked clinical improvement with phases of disease remission are observed with decreased exposure to allergens and irritants in stable dust [2]. However, complete avoidance of the offending agents is often impossible, and medical treatment with corticosteroids and bronchodilators becomes necessary to relieve acute asthma symptoms. Chronic changes and residual airway obstruction related to airway remodeling persist in disease remission [3]. sEA is currently incurable and its consequences range from substantial welfare concerns to negative economic effects on equestrian sports and horse breeding [4]. Elucidation of the underlying pathophysiological mechanisms of asthma in horses is needed to improve prevention and treatment of this condition. Importantly, such advances may also improve our understanding of asthma in humans, as immunological and pathophysiological characteristics of sEA show important parallels with allergic as well as severe neutrophilic human asthma [5].

Asthma in humans has a prevalence of around 4.3% globally, and not only poses a significant threat to human health, but also presents a massive economic burden [6,7]. Much of the pathophysiological mechanisms of asthma are studied in experimental animal models (mostly rodents). However, sEA offers one of the few spontaneous models of human asthma [4]. In order to improve the management of sEA in horses as well as discover novel comparative information relevant to human asthma, specific molecular pathways and mechanisms at the transcriptomic level need to be better understood. The current evidence regarding the immunological basis of sEA is conflicting. There are data that support both aspects of type I and type III hypersensitivities as well as involvement of the major immunological pathways including Th1, Th2, and Th17 [8,9,10,11]. An immediate-phase response is lacking in sEA, but affected horses characteristically show a delayed-phase allergic reaction to antigen challenge with neutrophil recruitment to the bronchial lumen and mixed Th-cell responses [12]. Several studies have reported a predominant Th2-type immune response (characterized by increased IL-4, IL-5, and IL-13 expression) [13,14,15] and evidence from genome-wide association studies (GWAS)tentatively supports *IL4R* as a possible candidate gene [16], while other investigations reported a mixed Th1–Th2-type response [17,18,19].

In human asthma, which is now recognized as a complex, heterogeneous disease group, the intricate and intertwined gene–environment interactions [20] are being categorized and assigned to sub-conditions with different clinical presentations (phenotypes) and diverse underlying pathophysiological mechanisms (endotypes) [5,21]. Recently, transcriptomic analyses of bronchial epithelium and of stimulated peripheral blood mononuclear cells (PBMC) have been performed in sEA [9,22]. Differentially expressed genes, pathways, and networks were identified in the airway epithelium of affected horses (e.g., MMP1, IL8, TLR4, and MMP9; neutrophil chemotaxis, immune, and inflammatory responses; circadian rhythm dysregulation; and the sonic Hedgehog pathway) that were also found to contribute to human asthma. Furthermore, novel differentially expressed genes (e.g., *CXCL13*, which is predominantly produced by Th17 cells) as well as dysregulated cellular pathways including immune cell trafficking, neutrophil chemotaxis, immune and inflammatory responses, cell cycle regulation, and apoptosis were discovered in allergen-stimulated peripheral blood mononuclear cells from horses with sEA [9]. MicroRNAs (miRNA) orchestrate many biological and developmental processes and the top differentially expressed serum miRNAs in horses suggest that dysregulated pathways are involved in airway remodeling in sEA (e.g., CD4+ T cell function, and a Th2/Th17-type immune response) [11].

The present study aimed to investigate miRNA and mRNA expression changes in an integrative analysis in the lung tissue of sEA-affected horses in exacerbation, in remission as well as in healthy controls. We report dysregulated miRNA-mRNA networks in sEA and interpret the findings in light of molecular mechanisms described in human asthma and experimental animal models.

## 2. Materials and Methods

### 2.1. Ethics Statement

This study used lung tissue biopsy samples that were archived in the Equine Respiratory Tissue Biobank (http://www.ertb.ca/) as a product of previous scientific studies [23]. All experimental procedures were performed in accordance with the Canadian Council for Animal Care guidelines and were approved by the Animal Care Committee for the Faculty of Veterinary Medicine of the University of Montreal (Rech-1324).

### 2.2. Animals, Sample Collection, and Phenotyping

The peripheral lung tissue samples analyzed in this study are a subset of samples published previously [23]. The samples were provided by the Equine Respiratory Tissue Biobank. Horses that were part of this study were medically examined by assessing lung function and bronchoalveolar lavage fluid (BALF) cytology before euthanasia. The horses were part of three cohorts: (1) sEA-affected horses in exacerbation; (2) sEA-affected horses in remission; and (3) healthy control horses. An overview of the characteristics and composition of the groups is indicated in Appendix A.

Inclusion criteria for control horses were the absence of a history of recurrent respiratory distress and other clinically apparent respiratory or systemic disorders at the time of euthanasia or the preceding six months. Additionally, control horses were required to show normal eosinophil, neutrophil, and mast cell levels during BALF cytology examination as well as a normal premortem lung function. A horse was identified as suffering from sEA only if it demonstrated repeated and reversible episodes of labored breathing at rest while not showing signs of systemic illness. In addition, an altered lung function, as well as more than 5% neutrophils during BALF cytology, was a requirement (details on inclusion criteria are listed in [23]).

Horses with sEA were categorized as in exacerbation or in remission based on the pre-mortem clinical examination of the lung function as well as the treatment history. Exacerbated horses were stabled for at least four weeks, fed hay, received no treatment, and showed increased pulmonary resistance and elastance. Remission horses were kept on pasture for at least four weeks without treatment and showed normalization of pulmonary resistance and elastance. After euthanasia and collection of the peripheral lung tissue sample, the samples were snap frozen in liquid nitrogen and stored at −80 °C in RNAlater (Thermo Fisher Scientific, Waltham, MA, USA) until use.

### 2.3. RNA Extraction and Sequencing

Homogenization of 30 mg of equine lung tissue was carried out using a TissueLyser instrument (Qiagen, Hilden, Germany) with an oscillation frequency of 25/s for 2 min.

Total RNA extraction was performed using the NucleoSpin miRNA kit (Macherey-Nagel, Düren, Germany) according to the manufacturer’s protocol. This extraction method allows collecting the isolated RNA in two separate fractions based on a size limit [one fraction containing small RNA <200 nt) and one fraction with large RNA (≥200 nt). The quality of all extracts (small and large RNA fraction for each sample) was assessed using a Bioanalyzer (Agilent, Santa Clara, CA, USA) device. The quantity of the extracts was determined with a Qubit fluorimeter 2.0 (Invitrogen, Carlsbad, CA, USA).

After having passed the quality and quantity assessments, the small RNA samples were converted into libraries using the TruSeq Small RNA Library Prep Kit (Illumina, San Diego, CA, USA) following the manufacturer′s protocol and using a size-selection range of 18 to 30 nt. Single-end sequencing of the small RNA libraries was carried out on a HiSeq 2500 system (Illumina) with 50 sequencing cycles. To obtain mRNA libraries, we used the RNA TruSeq Sample Preparation Kit v2 (Illumina), according to the manufacturer’s protocol. Next-generation sequencing was then carried out on a HiSeq 2500 device (Illumina) on two lanes with 2 × 150 bp paired-end sequencing. The obtained raw data files are available on Sequence Read Archive as Bioproject PRJNA639022.

### 2.4. Bioinformatic Analysis of miRNA Sequencing Data

To analyze the resulting next-generation sequencing dataset of the small RNA fraction containing the miRNA, we applied a bioinformatics pipeline, which was established during a previously published project [11]. Briefly, we assessed the raw sequencing data quality using the tool FastQC [24]. Removing adapters and trimming low-quality base calls (<q20) was carried out with the software cutadapt [25]. The cleaned sequencing reads were aligned to the equine reference genome EquCab3.0 [26] using the tool miRDeep2 (version 0.0.8) [27]. Subsequently, we detected novel equine miRNA based on the aligned sequencing reads. To filter low-confidence novel miRNAs, we applied a stringent miRDeep2 cutoff score of ≥5, as this maximized both the signal-to-noise ratio as well as the estimated number of true positives. The resulting set of newly detected putative high-quality novel equine miRNAs was then combined with the set of known equine miRNAs from the database miRBase (v.22) [28]. The combined set of equine miRNAs was used to quantify the miRNA expression levels in our sequencing data using the script quantifier.pl of the tool miRDeep2. To detect differentially expressed miRNAs between the cohorts, the R package DESeq2 [29] was used. We accounted for the covariates′ sex and RNA extraction batch in the linear model. miRNAs showing a false discovery rate adjusted *p*-value (P_adjusted_) determined by DESeq2 below a threshold of 0.05 were considered to be significantly differentially expressed. Pathway analysis was carried out using the tool mirPath v.3 using the homologous human miRNAs of the differentially expressed miRNAs [30].

### 2.5. Bioinformatic Analysis of mRNA Sequencing Data

The raw sequencing reads obtained after the next-generation sequencing of the mRNA libraries were quality controlled using the tool FastQC. Subsequently, adapters were trimmed, and low-quality bases (q < 20) were filtered using the software cutadapt. The cleaned sequencing reads were then aligned to the equine reference genome EquCab3.0 (NCBI Equus caballus Annotation Release 103: NC_009144.3 1 188260577) with the tool STAR (v.2.5.3a) [31]. Gene expression quantification was carried out using the tool htseq-count [32]. Differential expression analyses were carried out using DESeq2 and genes showing a *P*_adjusted_ ≤ 0.05 were considered to be statistically significantly differentially expressed. We accounted for the covariates sex and RNA extraction batch in the linear model. Downstream GO term enrichment was carried out using the tool PANTHER version 14.0 [33].

### 2.6. Network Analysis of Differentially Expressed miRNAs and mRNAs

As a first method to visualize, explore, and analyze the connection between differential expressed miRNA and mRNA genes, the software Cytoscape (v3.7.1) [34] was used in combination with the third-party application CyTargetLinker (v4.1.0) [35]. To construct the networks for all three cohort comparisons, we applied the same protocol. First, we imported the list of the significant miRNAs (their respective human homologs), then we expanded these miRNAs with the help of CyTargetLinker with all their target genes reported by two miRNA–mRNA interaction databases: TargetScan v.7.2 and miRTarBase v.7.0. Then, we filtered the resulting big network using the list of significantly differentially expressed genes for the corresponding comparison of two of the study cohorts. These analyses yielded a network depicting significantly differentially expressed miRNAs and their reported target genes that were also shown to be differentially expressed. Graphically, we chose to depict the expression levels in the form of the log2 fold changes as the fill color of the nodes, while we scaled the node size based on the number of outgoing and ingoing edges.

### 2.7. Integrative Negative Correlation Analysis of miRNA and mRNA Expression Profiles

To detect significantly negatively correlated expression values between miRNA and mRNA, we used the tool miRComb [36]. For this analysis, we only used differentially expressed miRNAs (only known miRNAs) and mRNAs from the three pairwise contrasts applied during differential expression analyses (exacerbation vs. controls, remission vs. controls, and exacerbation vs. remission). Then, we calculated a pairwise Pearson correlation coefficient of the DESeq2 normalized and log2 transformed expression values. Of these, we kept the significantly negatively correlated miRNA-mRNA pairs showing a negative Pearson correlation coefficient, and Benjamini–Hochberg corrected *p*-value ≤ 0.05. Additionally, to increase the confidence and biological significance of the obtained results, we requested the miRNA–mRNA interaction to be reported in humans by at least one of four databases (TargetScan, microCosm, miRDB, and miRSVR). 

### 2.8. Code Availability

All scripts used in the course of this study can be found on GitHub: www.github.com/MaHulliger/Lung/.

## 3. Results

### 3.1. Study Design

This study included eleven clinically phenotyped horses affected by sEA, of which six were in an exacerbated state and five were in clinical remission. The control group consisted of eight healthy horses. To investigate potentially dysregulated miRNA-mRNA networks underlying the pathology of sEA, we applied a comprehensive approach incorporating miRNA expression profiling and mRNA expression profiling, followed by an integrative miRNA-mRNA negative-correlation approach (Figure 1).

### 3.2. miRNA Expression Profiles and Novel miRNA Detection in Lung Tissue of Horses

In order to profile the miRNA expression levels in horses, we conducted a high-throughput small RNA sequencing experiment using sRNA extracts from peripheral lung tissue samples. After assessing the quality of the obtained data (Appendix A) and running the pipeline, we detected 278 putative high-confidence novel equine miRNAs. Table 1 lists the top 10 novel miRNAs according to miRDeep2 and the complete list of all putative novel miRNAs is available in Appendix A.

### 3.3. Differential miRNA Expression Analyses

The merged set of known equine miRNAs (*n* = 690) with the set of our putative novel high-confidence equine miRNAs (*n* = 278) were used to perform pairwise differential miRNA expression analyses between the three cohorts (Table 2). A collection of quality control figures for the miRNA differential expression analyses can be found in Appendix B (Figure A1, Figure A2, Figure A3, Figure A4, Figure A5, Figure A6 and Figure A7).

#### 3.3.1. Differential miRNA Expression Analysis: sEA-Affected Horses in Exacerbation versus Healthy Control Horses

Differential expression analysis of miRNAs between horses in an exacerbated state and healthy control horses yielded 8 miRNAs (Table 3). The miRNA showing the lowest *p*-value of 6.30 × 10^−5^ was eca-miR-142-3p with a log2 fold change of 0.69. In addition to the seven known miRNAs that showed a significant differential expression pattern, there was one putative novel equine miRNA that was significantly differentially expressed between the two groups: eca-miR-chrX_37753 (*P*_adjusted_ = 0.015).

#### 3.3.2. Differential miRNA Expression Analysis: sEA-Affected Horses in Remission versus Healthy Control Horses

Applying the next contrast of sEA-affected horses in remission versus healthy controls, we detected 11 significantly differentially expressed miRNAs. Of these, five were novel, while six where known equine miRNAs (Table 4). The miRNA with the lowest *p*-value of 2.52 × 10^7^ and the one with the highest log2 fold change (3.36) was a putative novel miRNA with the provisional ID eca-miR-NW_019643269.1_38788.

#### 3.3.3. Differential miRNA Expression Analysis: sEA-Affected Horses in Exacerbation Versus sEA-Affected Horses in Remission

The third and final differential miRNA expression analysis was carried out comparing affected horses in exacerbation against affected horses in remission. We identified four differentially expressed miRNAs (Table 5). The miRNA showing the lowest P_adjusted_ and the lowest negative log2 fold change (−2.6) again was the putative novel miRNA with the provisional ID eca-miR-NW_019643269.1_38788 with a P_adjusted_ of 6.00 × 10^−3^. The three remaining significantly differentially expressed miRNAs were eca-miR-146-5p, eca-miR-135b, and eca-miR-31.

Considering all three comparisons, a total of six putative novel equine miRNAs were detected to be significantly differentially expressed. While three of these miRNAs have a clear human homologous miRNA (eca-miR-chr15_8716, eca-miR-chr7_32350, and eca-miR-NW_019643269.1_38788), two novel miRNAs showed an identical seed sequence (nucleotides 2 to 8 at the 5′ end), but low percent identity to human miRNAs (eca-miR-chrX_37117 and eca-miR-chr6_31338). For the remaining miRNA (eca-miR-chrX_37753), the similarity to human miRNAs was very limited, however, a bovine homologous miRNA was reported (Table 6).

Figure 2 shows the Venn diagram comparing the potential overlaps of identified differential expressed miRNAs between the three cohorts.

### 3.4. Pathway Analysis of Differentially Expressed miRNAs and Their Target Genes

Pathway analysis using the human homologous miRNAs of the differentially expressed miRNAs of all three contrasts and experimentally confirmed target genes from Tarbase v7 that are potentially relevant for the molecular pathology of sEA yielded several significantly enriched pathways (e.g., TGF-β signaling, PI3K-Akt signaling, FoxO signaling, and T cell receptor signaling). Appendix A lists the results of these analyses in detail.

### 3.5. mRNA Expression Profiles in Lung Tissue of Horses and Differential Gene Expression Analyses

Detailed quality control of the raw mRNA sequencing data was carried out before applying the analytical pipeline (Appendix A). Analogous to the miRNA differential expression analyses, pairwise differentially expressed mRNAs between the three conditions were identified (Table 7). A set of quality control figures can be found in Appendix B (Figure A8, Figure A9, Figure A10 and Figure A11).

#### 3.5.1. Differential mRNA Expression Analysis: sEA-Affected Horses in Exacerbation Versus Healthy Control Horses

Applying the first contrast, comparing asthmatic horses in exacerbation to healthy controls, we identified a total of 274 significantly differentially expressed genes (Table 8, Appendix A). Of the significant genes, 169 were upregulated and 105 genes were determined to be downregulated in horses in an exacerbated state compared with control horses.

The upregulated differentially expressed genes showed a mean log2 fold change of +5.4 (±0.4 sd) and the downregulated differentially expressed genes exhibited a mean log2 fold change of −3.67 (±0.90 sd). Of these differentially expressed genes, 46 have previously been associated with asthma in humans (according to disGeNET v6.0; Appendix A).

#### 3.5.2. Differential mRNA Expression Analysis: sEA-Affected Horses in Remission Versus Healthy Control Horses

The second comparison covering asthmatic horses in remission and healthy control horses yielded a total of 53 significantly differentially expressed (Table 9, Appendix A). Of these, 32 were upregulated in horses in remission relative to healthy control horses and 21 genes were downregulated. The upregulated differentially expressed genes showed a mean log2 fold change of 1.20 (±0.77 sd), while the downregulated differentially expressed genes presented a mean log2 fold change of −1.09 (±0.50 sd).

#### 3.5.3. Differential mRNA Expression Analysis: sEA-Affected Horses in Exacerbation versus sEA-Affected Horses in Remission

Finally, we performed a differential gene expression analysis for the last contrast between the three groups: asthmatic horses in an exacerbated state against asthmatic horses in remission. This analysis yielded a total of 257 significantly differentially expressed genes between the two groups (Table 10, Appendix A). A total of 180 genes were significantly upregulated in horses in an exacerbated state relative to horses in remission, while 77 genes were significantly downregulated. The observed mean log2 fold change of the significantly upregulated genes was 2.95 (±2.26 sd), while the mean log2 fold change of the downregulated genes was −0.56 (±0.51 sd).

To investigate potential overlaps of the three sets of significantly differentially expressed genes, we created a visualization of the overlaps in the form of a Venn diagram (Figure 3).

### 3.6. Gene Ontology (GO)-Term Enrichment Analysis of Differentially Expressed mRNAs

To check functional enrichment of the reported sets of differentially expressed genes, we used PantherDB to perform gene ontology (GO) term enrichment analysis. The enriched GO-terms for the comparison between horses in an exacerbated state and control horses mostly spanned inflammatory processes and pathways related to tissue remodeling (Figure 4). The significant GO-term with the lowest adjusted *p*-value was “inflammatory response”, with 18 differentially expressed genes involved. For the analysis comparing horses in an exacerbated state to horses in remission, the significantly enriched GO-terms mostly represented stress response and developmental pathways (Figure 4). The GO-term with the lowest adjusted *p*-value was “response to stress”, including 44 differentially expressed genes. Owing to the comparably low number of differentially expressed genes in the last contrast comparing horses in remission to healthy horses, no GO-terms were detected to be significantly enriched.

### 3.7. Network Analyses of Differentially Expressed miRNAs and mRNAs

In order to visualize the reported sets of differentially expressed miRNAs and mRNAs, we constructed a network for all three pairwise comparisons. We used two miRNA–mRNA interaction databases (TargetScan and miRTarBase). For the network analysis, we only included differentially expressed genes that were reported to be targets of at least one differentially expressed miRNA. With this approach, we aimed to investigate potentially dysregulated miRNA-mRNA networks that might underlie the pathophysiology of sEA.

#### 3.7.1. Network Analysis: sEA-Affected Horses in Exacerbation versus Healthy Control Horses

The first constructed network (exacerbation vs. controls) showed miR-142-5p and miR-26a-5p as major hubs regulating several mRNAs (Figure 5). The network shows several genes regulated by more than one miRNA (*CCNT2, SCD, MEX3A, FRMPD4, SYDE2, RGS4, GRB10, TMEM260, ST3GAL6, HTR2A*, and *HAS3*). These genes might be an interesting starting point for future research into asthma exacerbation. To study this synergistic network in more detail, we expanded the resulting network with transcription factors interacting with genes in the network (Appendix A) and annotated it with biological pathways (Appendix A). This analysis revealed, for example, that the transcription factors STAT3 (which is a key transcription factor in Th17 cellular development [37]), GATA2, and NFKB1 might play an important role in this molecular network. Additional potentially affected biological pathways detected were the following: IL-4 signaling pathway, IL-17 signaling pathway, TGF-β signaling, NOTCH signaling, and the development of pulmonary dendritic cell and macrophage subsets.

#### 3.7.2. Network Analysis: sEA-Affected Horses in Remission versus Healthy Control Horses

The second network from the comparison of sEA-affected horses in remission to healthy controls was rather limited, because of the lower number of significant mRNAs detected (Figure 6). However, *ARID1B*, regulated by two differentially expressed miRNAs, presents itself as a potential target for investigating the molecular mechanism of the long-term response in asthmatic horses in remission.

#### 3.7.3. Network Analysis: sEA-Affected Horses in Exacerbation versus sEA-Affected Horses in Remission

The third and final constructed network indicated a potential prominent role for the highly overexpressed *ST6GAL2* in resolution of acute airway inflammation (as this covers the comparison of asthmatic horses in exacerbation to asthmatic horses in remission). Interestingly, most target genes in the network are overexpressed in asthmatic horses in exacerbation (Figure 7). We also expanded this resulting network with transcription factor–gene interactions (Appendix A) and biological pathways (Appendix A).

### 3.8. Identification of Putative miRNA-mRNA Regulatory Interactions Associated with sEA

#### 3.8.1. Identification of Negatively Correlated miRNA-mRNA Pairs: sEA-Affected Horses in Exacerbation versus Healthy Control Horses

For the first contrast, we calculated pairwise Pearson’s correlation between all possible combinations of the 274 differentially expressed (DE) genes, and the seven differentially expressed miRNAs (novel miRNAs were not used for these analyses). Applying this approach, we were able to detect 14 significantly negatively correlated miRNA-mRNA pairs showing a mean Pearson’s correlation coefficient of −0.75 (±0.08 sd) (Table 11).

#### 3.8.2. Identification of Negatively Correlated miRNA-mRNA Pairs: sEA-Affected Horses in Remission versus Healthy Control Horses

Applying the same pipeline to the 53 differentially expressed genes and six known differentially expressed miRNAs detected between asthmatic horses in remission and healthy control horses revealed four miRNA–mRNA pairs that were significantly negatively correlated. eca-miR-142-3p showed negatively correlated expression values with three putative target genes: *ADAM22, GORAB*, and *NR3C2*. The mean Pearson’s correlation coefficient obtained for all detected pairs was −0.78 (±0.09 sd) (Table 12).

#### 3.8.3. Identification of Negatively Correlated miRNA-mRNA Pairs: sEA-Affected Horses in Exacerbation Versus sEA-Affected Horses in Remission

For the last contrast analyzed, we used the 257 differentially expressed genes and the three differentially expressed known miRNAs of the comparison between asthmatic horses in an exacerbated state and asthmatic horses in remission. This analysis revealed two significant negatively correlated miRNA–mRNA pairs: eca-miR-146b-5p with *OSGIN2*, as well as the same miRNA with the gene *UBTD2* with a Pearson’s correlation coefficient of −0.82 (Table 13).

## 4. Discussion

In this study, we identified molecular mechanisms and pathways that may play important roles in the pathogenesis of sEA and offer novel comparative data for human asthma research based on comprehensive global miRNA and mRNA expression profiling in lung tissue samples of sEA-affected horses in exacerbation, in remission, and in healthy controls. To the best of our knowledge, this is the first study using an integrative approach combining miRNA and mRNA expression profiles in a spontaneous model of asthma. Dysregulated miRNA-mRNA networks were identified, revealing novel and striking similarities at the transcriptomic level between sEA and human asthma pheno- and endotypes (particularly severe neutrophilic asthma) based on in-depth literature research (Appendix A).

Some of the most prominent findings relate to inflammatory processes documented in severe neutrophilic asthma of humans. We decided to focus on the three miRNAs (miR-142-3p, miR-142-5p, and miR-223) that were found to be upregulated in both comparisons, asthmatic horses in exacerbation versus healthy controls and horses in remission versus controls (Figure 2). Two of these miRNAs, namely miR-142-3p and miR-223, play a central role in the molecular pathology of human severe neutrophilic asthma by promoting airway inflammation and obstruction [38]. It was not only shown that these two miRNAs are overexpressed in the sputum of humans with severe neutrophilic asthma, but also that their overexpression is strongly associated with inflammatory parameters, airway obstruction, and proinflammatory cytokine levels [38]. Validated target genes of these miRNAs are implicated in several inflammatory pathways including Toll-like receptor signaling, NOD-like receptor signaling, and mitogen-activated protein kinase (MAPK) signaling, leading to increased levels of IL-1β, IL-6, and IL-8 [38]. There was a significant increase in the corresponding IL-1β and IL-8 levels in patients with severe asthma [38]. Furthermore, the expression levels were correlated with the neutrophil counts in the sputum samples. Several studies reporting increased expression levels of IL-1β and IL-8 in humans with neutrophilic asthma agree with these results [39,40]. Interestingly, these pathways and signature genes also are consistent with findings in horses with sEA, including recent transcriptomic analyses of the bronchial epithelium of affected horses relative to healthy horses in response to antigen challenge [22] and several other studies reporting overexpressed IL-1β and IL-8 [17,41,42,43,44].

Although the exact molecular mechanism of how these two miRNAs affect the pathophysiology of severe neutrophilic asthma remains unknown, recent findings point towards their role in the development of other important disorders: in the lungs of tuberculosis patients, miR-223 upregulates miR-142-3p expression as part of the ‘miR-223-CEBP-β-LMO2-miR-142’ pathway [45] and controls neutrophil recruitment as well as neutrophil-driven inflammation [39]. Furthermore, miR-223 plays an essential role in autoimmune inflammation, where it mediates the myeloid dendritic cell induced pathologic Th17 response [46]. Its targeting of the inflammatory mediator ICAM1 [47] and its effect on macrophage polarization towards the M2 type [48] suggest additional important functions of miR-223 in neutrophilic asthma. Furthermore, miR-142-3p is implicated in aberrant WNT signaling during airway remodeling [49], in pro-inflammatory processes via monocyte-derived dendritic cells (while suppressing Treg expansion) [50], and in the suppression of Th1 cytokines by dendritic cells [51]. Additionally, miR-142-3p is overexpressed in Lipopolysaccharide (LPS)-stimulated macrophages [52] and impairs antigen processing [53]. Thus, dysregulated miR-223 and miR-142-3p might affect sEA pathophysiology by promoting airway remodeling, positively regulating neutrophil recruitment and airway inflammation, and possibly affecting macrophage polarization. MiR-223, together with miR-31, showed an interesting pattern of a rather high within-group variation for horses in exacerbation (Appendix B
Figure A5). While this might be expected owing to the symptomatic stage of the disease, it might also reflect the presence of potential unidentified endotypes of sEA.

Several differentially expressed miRNAs detected in the lungs of horses in exacerbation compared to controls are linked with the positive regulation of the Th17 pathway. Indeed, downregulated miR-26a leads to increased expression of IL-17 and IL-6 [54] and affects Th17/Treg balance towards Th17 cells [55]. Additionally, miR-31 drives Th17 cell differentiation through targeting IL-25, while suppressing Treg differentiation through FOXP3 [56]. Moreover, miR-212 promotes Th17 cell differentiation likely by targeting Bcl-6 [57], while miR-223 regulates the Th17 immune response in a myeloid dendritic cell-driven manner [46]. The potential involvement of an increased Th17-type immune response is further supported by multiple dysregulated miRNAs in the second contrast comparing sEA-affected horses in remission to healthy horses (miR-363, miR-379, miR-193a-3p, miR-135b [58,59,60,61,62]). Th17 cells recruit neutrophils through releasing Th17 cytokines (IL-17A, IL17F, and IL-22) and epithelial-derived neutrophilic chemokines [63]. Interestingly, the miRNA–mRNA networks that we expanded with regulatory transcription factor interactions support the hypothesis of a dysregulated Th17 cell differentiation pathway (Appendix A). The expanded network suggests an important role for the IL-17 signaling pathway and for STAT3, which is a key transcription factor in Th17 cellular development. Taken together, these findings suggest an important role of the Th17 pathway in the pathophysiology of sEA, especially in disease exacerbation. The dysregulated miRNAs presented in this study might affect sEA pathology by leading to increased recruitment and differentiation of Th17 cells in the lung and increased Th17 cell differentiation. The Th17-mediated inflammation then in turn potentially leads to increased neutrophilic airway inflammation and increased airway remodeling [63]. This agrees with our previous findings in stimulated PBMC and those of other groups in mediastinal lymph nodes, BALF, and epithelial cells of sEA-affected horses [9,11,64,65,66]. However, previous findings in sEA also suggest that Th2-type immune responses consistent with allergic-type asthma play a role in sEA [12,13,14], and some of our present results support this pathway.

Differential gene expression analysis showed hundreds of genes to be significantly differentially expressed between the three group comparisons. *IL4R* was among the significantly differentially expressed genes of the first comparison (exacerbation against controls) and is also on the list of genes previously associated with human asthma (Appendix A). *IL4R*, which codes for the cytokine receptor binding the Th2 cytokines IL-4 and IL-13, was previously identified by our group as a candidate gene for sEA in a GWAS experiment [67]. Our network analysis, expanded with annotated pathways (Appendix A), also supports involvement of the IL-4 signaling pathway in sEA. Of note, our integrative analysis of miRNA–mRNA pairs further substantiates the role of the Th2-type response during sEA exacerbation. Carbonic anhydrase II (*CA2*), which was negatively correlated with miR-26a-5p and significantly upregulated in exacerbated horses, is known to be upregulated by Th2 cytokines like IL-4 and IL-13, and reportedly promotes airway hyperresponsiveness [68,69]. Moreover, RGS4 (also negatively correlated with miR-26a-5p) plays a role in modulating airway hyper-responsiveness and airway obstruction [70,71].

Our results suggest additional dysregulated processes in sEA exacerbation including alternative macrophage activation (miR-26a, miR-142-5p, miR-212, miR-223 [48,52]) and NF-kB signaling (miR-224 [72]). The latter being additionally supported by our expanded transcription factor–gene interaction network analysis (Appendix A) and by earlier studies investigating sEA [11,73,74]. Importantly, our miRNA–mRNA integrative analysis also indicates that NF-kB plays an important role in sEA: *CYP1A1* was negatively correlated with miR-142-3p. This NF-kB controlled gene has been associated with asthma and lung function deficits [75,76]. Additionally, SIAH-1 (targeted by miR-142-3p) regulates NF-kB activity via TNF-α and has been shown to interact with many other genes involved in the mammalian immune response [77].

Counteracting these deleterious processes, we also identified anti-inflammatory pathways taking place during asthma exacerbation, but not in remission. MiR-146-5p targets inflammatory mediators like *COX2, IL-1β, KIT, NFKB1, TLR4, TRAF6*, and *UHRF1* [78] and has been proposed as a predictor for asthma exacerbations in childhood asthma [79,80]. The anti-inflammatory miR-135b acts by targeting *FOXN3* and *RECK*, an inhibitor of MMP2, and the known sEA pathogenesis factor MMP9 [81,82,83]. These anti-inflammatory actions might be the immunological counterbalance, and thus the defense mechanism of the lung against acute inflammation (likely Th17-mediated, as outlined above) during disease exacerbation.

Paralleling the pathology of severe human asthma [84], the significantly dysregulated miRNAs (miR-363, miR-379, miR-193a-3p) found in sEA-affected horses in remission (when compared with healthy horses) are likely to enhance TGF-β signaling [58,59,60]. This is further supported by the expanded network analysis including annotated pathways (Appendix A). Thus, these miRNAs may be key players in airway remodeling during disease remission, prolonging and amplifying effects of miRNAs that were significantly upregulated in exacerbation. Specifically, miR-26a, miR-142-3p, and miR-142-5p promote airway remodeling (in parts by positively affecting TGF-β pathways) [55,85].

Besides the miRNA–mRNA pairs discussed above, our integrative miRNA–mRNA expression analysis revealed numerous negatively correlated pairs implicated in sEA. Many of these have not yet been investigated regarding a potential role in asthma pathology. Because we detected the miRNA–mRNA pair *ADAM22*-miR-142-3p in both comparisons (exacerbation vs. controls and remission vs. controls), it might be a key player in the pathophysiology of sEA that not only affects exacerbation of the disease, but also disease remission. Additionally, we constructed miRNA–mRNA networks as an alternative approach. This visual representation helped to put some of the negatively correlated miRNA–mRNA pairs in a bigger perspective. Besides the pair *ADAM22*-miR-142-3p, the genes *CCNT2* (paired with miR-142-5p and miR-142-3p), *SIAH1*, *RGS4* (negatively correlated with miR-26a-5p), and *SYDE2* (paired with miR-31-5p) represent promising starting points for future research to validate the role of these dysregulated pathways in functional experiments.

Additionally, future research could be applied to validate our findings in an independent and larger study cohort, as well as by applying different technical approaches (e.g., RT-qPCR). While differential expression analyses with six or more samples per group are considered robust [86], an increased sample size can increase statistical power, and thus the ability to detect more differentially expressed genes and miRNAs.

In conclusion, we found important similarities between sEA (at the transcriptomic level) and severe neutrophilic asthma in humans. Specifically, our findings suggest that the dysregulated miRNAs and mRNAs might lead to a predominantly Th17-mediated immune response in exacerbation and remission of sEA. This Th17-mediated inflammation could potentially explain the prominent role of neutrophils in sEA. Other main findings also show parallels with allergic asthma (Th2-type response), and present potential pathways that promote airway remodeling. These results expand our understanding of the molecular mechanisms that underlie airway hyper-responsiveness and bronchospasm during disease exacerbation and airway remodeling during disease remission.

## Figures and Tables

**Figure 1 genes-11-01143-f001:**
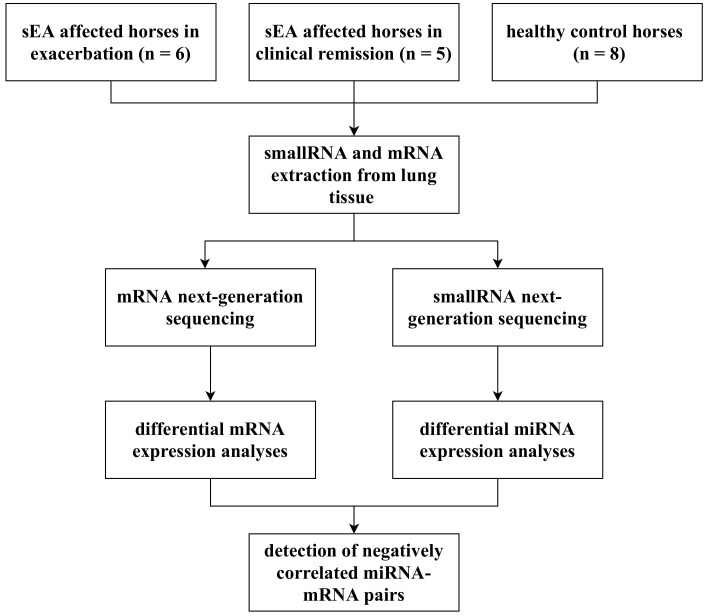
A schematic illustration of the workflow of this study. sEA, severe equine asthma.

**Figure 2 genes-11-01143-f002:**
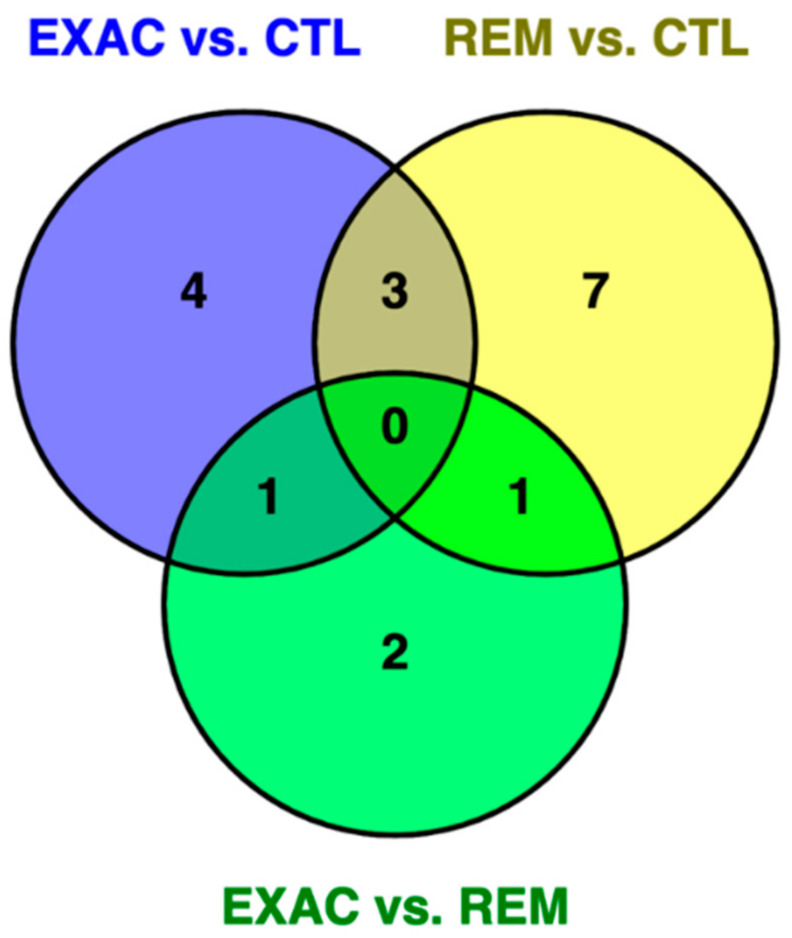
Venn diagram of differentially expressed miRNAs of the three comparisons: exacerbation (EXAC) vs. control (CTL), remission (REM) vs. CTL, and EXAC vs. REM. Shared between EXAC vs. CTL and EXAC vs. REM was eca-miR-31, while eca-miR-NW_019643269.1_38788 was shared between REM vs. CTL and EXAC vs. REM. Three miRNAs were shared by two comparisons (exacerbation vs. control and remission vs. control): eca-miR-142-3p, eca-miR-142-5p, and eca-miR-223.

**Figure 3 genes-11-01143-f003:**
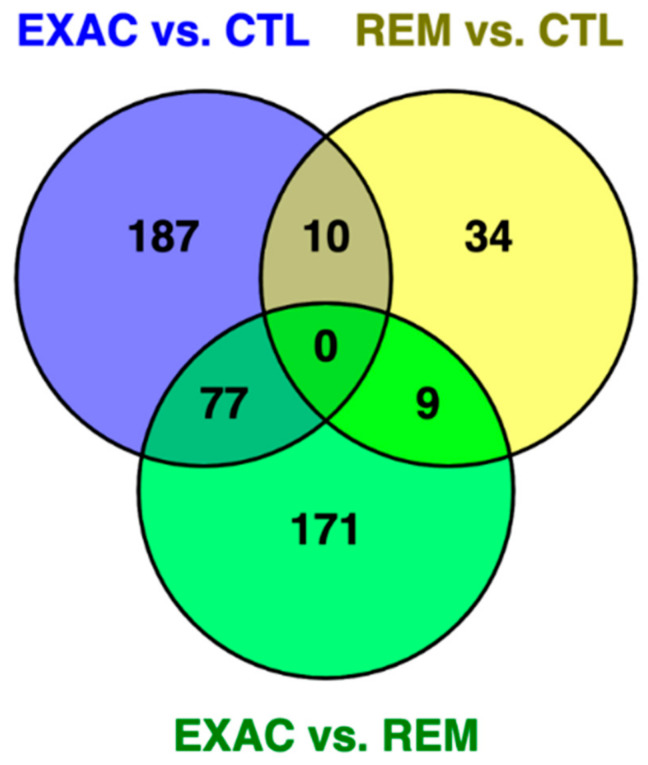
Venn diagram of the three sets of differentially expressed genes between the three comparisons: ‘exacerbation (EXAC) vs. controls (CTL)’, ‘remission (REM) vs. controls’, and ‘exacerbation vs. remission’. The three overlapping sets shared between two contrasts varied from 9 to 77 genes (Appendix A).

**Figure 4 genes-11-01143-f004:**
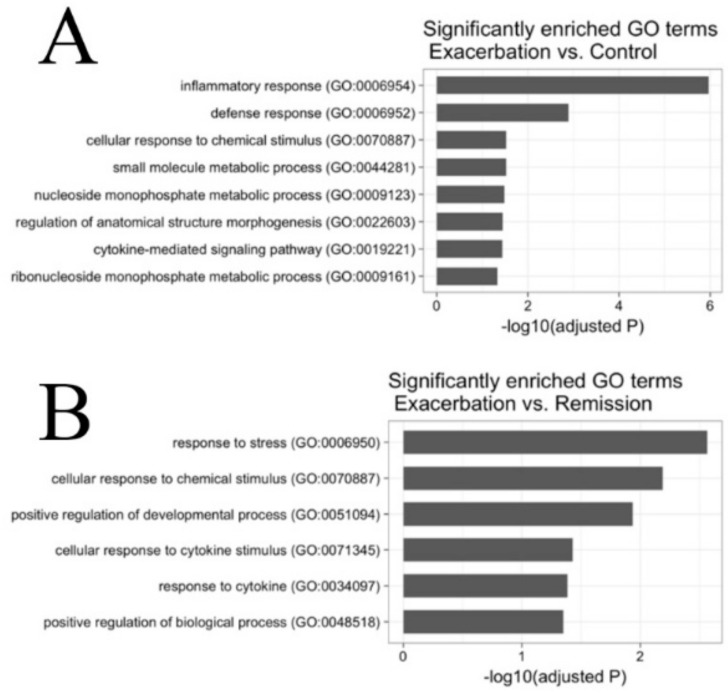
Gene ontology (GO)-term enrichment analyses indicating the adjusted *p*-value of the enriched terms. (**A**) Enriched GO-terms using differentially expressed genes between horses in an exacerbated state and control horses. (**B**) Enriched Go-terms using differentially expressed genes between horses in an exacerbated state and horses in remission.

**Figure 5 genes-11-01143-f005:**
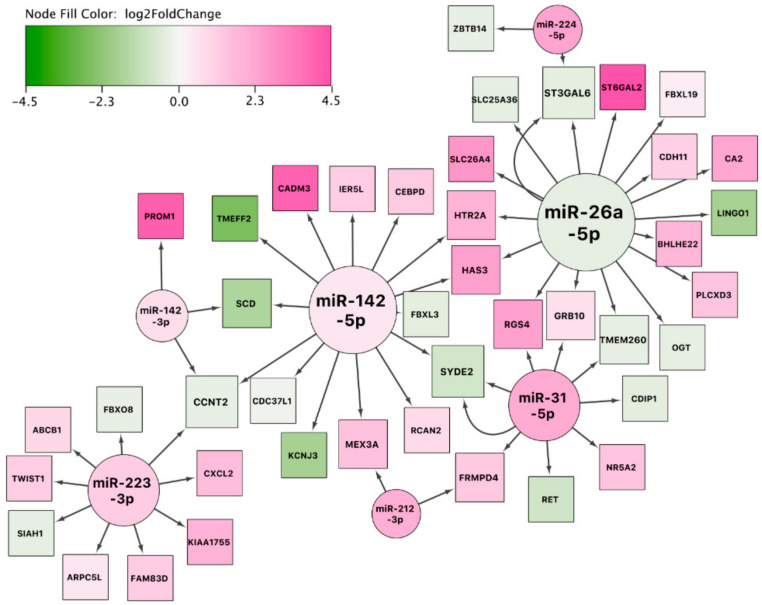
miRNA-gene interaction network using the set of differentially expressed miRNAs and mRNAs between asthmatic horses in exacerbation and healthy controls. The color of the interior of the nodes represents the reported log2 fold change of the miRNA or gene, the node shape represents the feature type (miRNA: circle, target gene: square), and the node size represents the degree (number of in- and outgoing edges). Two arrows between two nodes indicate that the interaction is supported by both databases (e.g., between miR-31-5p and SYDE2).

**Figure 6 genes-11-01143-f006:**
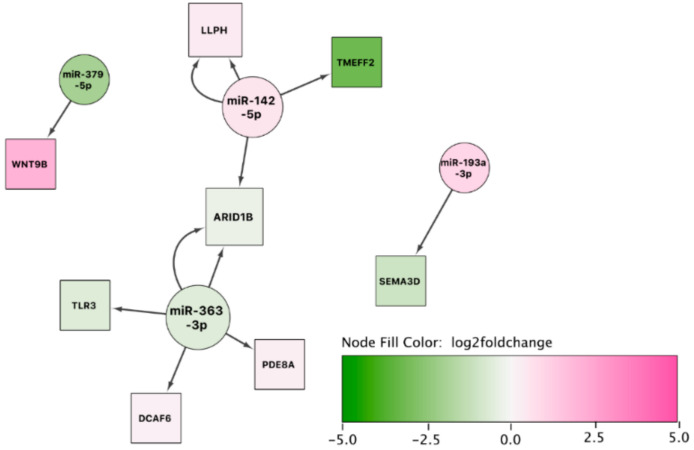
miRNA-gene interaction network using the set of differentially expressed miRNAs and mRNAs between asthmatic horses in remission and healthy controls. The color of the interior of the nodes represents the reported log2 fold change of the miRNA or gene, the node shape represents the feature type (miRNA: circle, target gene: square), and the node size represents the degree (number of in- and outgoing edges).

**Figure 7 genes-11-01143-f007:**
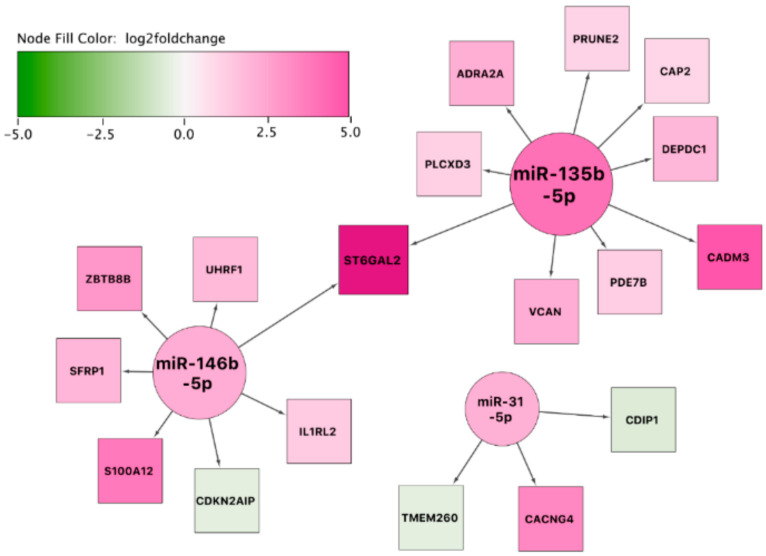
miRNA-gene interaction network using the set of differentially expressed miRNAs and mRNAs between affected horses in exacerbation and remission. The color of the interior of the nodes represents the reported log2 fold change of the miRNA or gene, the node shape represents the feature type (miRNA: circle, target gene: square), and the node size represents the degree (number of in- and outgoing edges).

**Table 1 genes-11-01143-t001:** Table of the top 10 putative novel equine miRNAs detected in this dataset ranked according to decreasing probability of the novel miRNA to be a true positive (represented by the miRDeep2 score; Appendix A). Additionally, the number of mismatches and the length of the putative novel miRNAs in nucleotides, as well as the mature consensus sequences, are indicated.

Provisional Name of Novel miRNA	Human Homologous miRNA ID	Number of Mismatches between Human/Equine miRNA	Consensus Mature Sequence
chr6_31290	hsa-let-7a-5p	0/21	ugagguaguaguuugugcuguu
chr20_16763	hsa-miR-30e-3p	2/20	uguaaacacccgacuggaagcc
chrX_37753	-	-	agagguaaaaaauugauuugacu
chr6_30538	hsa-miR-26b-5p	0/21	uucaaguaauucaggauagguu
chr6_31452	hsa-miR-375-5p	0/22	uuuguucguucggcucgcguga
chr24_19655	hsa-miR-203a-3p	0/22	gugaaauguuuaggaccacuaga
chr2_24882	hsa-miR-320a-3p	0/22	aaaagcuggguugagagggcga
NW_019643269.1_38788	-	-	gccgaucgaaagggagucgg
chr13_5604	hsa-miR-339-5p	0/22	ucccuguccuccaggagcucacu
chr7_33613	hsa-miR-181c-5p	0/22	aacauucaaccugucggugagu

**Table 2 genes-11-01143-t002:** Number of statistically significantly differentially expressed miRNAs in all three combinations of the three cohorts studied.

	Exacerbation vs. Controls	Remission vs. Controls	Exacerbation vs. Remission
Number of differentially expressed miRNAs (*P*_adjusted_ ≤ 0.05)	8	11	4

**Table 3 genes-11-01143-t003:** Differentially expressed miRNAs comparing horses in an exacerbated state and control horses. The mean DESeq2 normalized read counts, the log2 fold change, and the adjusted *p*-values are indicated.

miRNA ID	Mean Normalized Read Counts	Log2 Fold Change	P_adjusted_
eca-miR-142-3p	7016	0.69	6.30 × 10^−5^
eca-miR-26a	576,062	−0.38	8.70 × 10^−3^
eca-miR-142-5p	107,984	0.51	1.00 × 10^−2^
eca-miR-31	561	2.11	1.20 × 10^−2^
eca-miR-212	17	2	1.50 × 10^−2^
eca-miR-223	5088	1.16	1.50 × 10^−2^
eca-miR-224	10	2.28	1.50 × 10^−2^
eca-miR-chrX_37753	37,418	0.40	1.50 × 10^−2^

**Table 4 genes-11-01143-t004:** Differentially expressed miRNAs comparing asthmatic horses in remission and control horses. The mean DESeq2 normalized read counts, the log2 fold change, and the adjusted *p*-values are indicated.

miRNA ID	Mean Normalized Read Counts	Log2 Fold Change	P_adjusted_
eca-miR-NW_019643269.1_38788	3512	3.36	2.52 × 10^−7^
eca-miR-142-3p	7016	0.63	1.00 × 10^−3^
eca-miR-142-5p	107,984	0.59	2.00 × 10^−3^
eca-miR-chr15_8716	20	−2.57	2.00 × 10^−3^
eca-miR-chr7_32350	2186	−2.01	7.00 × 10^−3^
eca-miR-chrX_37117	790	−1.76	7.00 × 10^−3^
eca-miR-363	453	−0.61	2.70 × 10^−2^
eca-miR-379	16	−1.97	2.70 × 10^−2^
eca-miR-193a-3p	430	1.02	3.00 × 10^−2^
eca-miR-chr6_31338	323	−1.53	3.00 × 10^−2^
eca-miR-223	5093	1.04	4.60 × 10^−2^

**Table 5 genes-11-01143-t005:** Differentially expressed miRNAs comparing asthmatic horses in exacerbation and asthmatic horses in remission. The mean DESeq2 normalized read counts, the log2 fold change, and the adjusted *p*-values are indicated.

miRNA ID	Mean Normalized Read Counts	Log2 Fold Change	P_adjusted_
eca-miR-NW_019643269.1_38788	3512	−2.6	6.00 × 10^−3^
eca-miR-146b-5p	7515	2.3	9.00 × 10^−3^
eca-miR-135b	14	4.08	4.60 × 10^−2^
eca-miR-31	560	2.17	4.60 × 10^−2^

**Table 6 genes-11-01143-t006:** Differentially expressed putative novel equine miRNAs in any of the three group comparisons. The miRDeep2 given provisional miRNA ID, the consensus sequence of the novel equine miRNA, as well as a potential human homologous miRNA ID are listed.

Provisional miRNA ID	Consensus Sequence	Homologous miRNA	Percent Identity
eca-miR-NW_019643269.1_38788	gccgaucgaaagggagucgg	hsa-miR-5006-3p	93.8% (identical seed)
eca-miR-chr15_8716	accuggggaucugaggagg	hsa-miR-6852-5p	93.8% (identical seed)
eca-miR-chr7_32350	aucccaccacugccacca	hsa-miR-1260a	94.4% (identical seed)
eca-miR-chrX_37117	uuccccggcaucuccucca	hsa-miR-6763-3p	42.1% (identical seed)
eca-miR-chr6_31338	uccccggcuccuccacca	hsa-miR-4707-5p	38.9% (identical seed)
eca-miR-chrX_37753	agagguaaaaaauugauuugacu	bta-miR-6119-5p	100% (identical seed)

**Table 7 genes-11-01143-t007:** Number of statistically significantly differentially expressed genes in all three combinations of the three cohorts studied.

	Exacerbation vs. Controls	Remission vs. Controls	Exacerbation vs. Remission
Number of differentially expressed genes (*P*_adjusted_ ≤ 0.05)	274	53	257

**Table 8 genes-11-01143-t008:** Top 20 significantly differentially expressed genes between asthmatic horses in exacerbation and healthy horses according to the P_adjusted_. The normalized mean expression values calculated by DESeq2, the log2 fold change, as well as the adjusted *p*-values are listed.

Gene Name	Normalized Mean Expression Value	Log2 Fold Change	P_adjusted_
*HSPB7*	374	2.60	3.26 × 10^−7^
*LOC100056936*	814	4.33	2.53 × 10^−6^
*LOC100056698*	1050	3.71	3.65 × 10^−6^
*GSTA1*	321	5.73	4.41 × 10^−6^
*CXCL6*	840	5.23	6.63 × 10^−6^
*PMEPA1*	1042	0.81	7.73 × 10^−6^
*S100A12*	2715	4.09	4.58 × 10^−5^
*IGFBP3*	1063	4.02	7.04 × 10^−5^
*LOC100630497*	73	−1.39	9.63 × 10^−5^
*CADM3*	297	4.04	1.18 × 10^−4^
*AP2A1*	1463	0.38	1.46 × 10^−4^
*SAA1*	2773	2.70	1.46 × 10^−4^
*SCN1B*	106	1.64	1.46 × 10^−4^
*S100G*	41	−1.71	3.00 × 10^−4^
*LOC111768320*	74	1.42	3.06 × 10^−4^
*PTPRQ*	30	−2.58	3.27 × 10^−4^
*LOC111773141*	10	2.63	5.07 × 10^−4^
*PCDH15*	35	2.64	5.07 × 10^−4^
*IL1R2*	159	3.68	5.76 × 10^−4^

**Table 9 genes-11-01143-t009:** Top 20 significantly differentially expressed genes between asthmatic horses in remission and healthy horses according to the adjusted *p*-value. The normalized mean expression values calculated by DESeq2, the log2 fold change, as well as the adjusted *p*-values are listed.

Gene Name	Normalized Mean Expression Value	Log2 Fold Change	P_adjusted_
*YBX3*	3635	0.90	2.79 × 10^−5^
*TLR3*	612	−0.62	1.86 × 10^−4^
*LOC111771691*	63	0.80	3.62 × 10^−4^
*TMEFF2*	41	−3.23	3.62 × 10^−4^
*CXHXorf21*	100	−1.03	6.87 × 10^−4^
*EFHD1*	86	2.36	8.87 × 10^−4^
*RPL35*	3594	0.66	4.08 × 10^−3^
*MX2*	404	−1.1	4.12 × 10^−3^
*NR3C2*	839	−0.63	5.87 × 10^−3^
*POLG2*	325	0.93	5.87 × 10^−3^
*DDX60*	1838	−0.94	6.44 × 10^−3^
*F2RL2*	1480	0.6	6.44 × 10^−3^
*MAPKAP1*	854	0.37	6.44 × 10^−3^
*TTC19*	491	0.40	6.44 × 10^−3^
*URI1*	1477	0.37	7.78 × 10^−3^
*ADM2*	80	−0.91	8.08 × 10^−3^
*LOC102147438*	100	0.7	8.08 × 10^−3^
*DACH2*	153	−1.8	8.73 × 10^−3^
*TRIM17*	82	1.19	8.73 × 10^−3^

**Table 10 genes-11-01143-t010:** Top 20 significantly differentially expressed genes between asthmatic horses in an exacerbated state and asthmatic horses in remission according to the adjusted *p*-value. The normalized mean expression values calculated by DESeq2, the log2 fold change, as well as the adjusted *p*-values are listed.

Gene Name	Normalized Mean Expression Value	Log2 Fold Change	P_adjusted_
*LOC100630497*	73	−1.94	1.75 × 10^−7^
*CADM3*	297	4.85	6.37 × 10^−5^
*FOXL1*	77	1.27	6.37 × 10^−5^
*LOC111771691*	63	−0.95	6.37 × 10^−5^
*TFF1*	3983	9.14	1.95 × 10^−4^
*LOC106781592*	6117	8.91	2.13 × 10^−4^
*CXCL6*	840	5.06	6.16 × 10^−4^
*LOC111770812*	41	3.56	6.16 × 10^−4^
*LOC111771553*	194	−0.89	6.16 × 10^−4^
*TNF*	130	−1.16	6.16 × 10^−4^
*A2ML1*	1662	5.32	8.43 × 10^−4^
*IGFBP3*	1063	4.15	8.43 × 10^−4^
*TMEM260*	1562	−0.46	8.43 × 10^−4^
*ATP8B1*	1301	1.01	9.97 × 10^−4^
*IL1R2*	159	4.16	1.03 × 10^−3^
*CCL19*	103	2.37	1.25 × 10^−3^
*TCN1*	599	7.24	1.25 × 10^−3^
*ST6GAL2*	447	6.09	1.55 × 10^−3^
*LOC111768320*	74	1.5	1.80 × 10^−3^

**Table 11 genes-11-01143-t011:** This table displays the significantly negatively correlated miRNA-mRNA pairs for the comparison of horses in exacerbation and control horses. Pearson’s correlation coefficient and P_adjusted_ are indicated.

DE miRNA	DE mRNA	Pearson’s Correlation Coefficient	P_adjusted_
miR-31-5p	*SYDE2*	−0.91	4.00 × 10^−4^
miR-26a-5p	*RGS4*	−0.88	8.00 × 10^−4^
miR-224-5p	*FBXL3*	−0.82	2.80 × 10^−3^
miR-26a-5p	*CA2*	−0.78	6.20 × 10^−3^
miR-142-3p	*CCNT2*	−0.77	6.60 × 10^−3^
miR-142-3p	*ADAM22*	−0.76	7.90 × 10^−3^
miR-142-3p	*CYP1A1*	−0.73	1.20 × 10^−2^
miR-142-3p	*PPM1K*	−0.71	1.50 × 10^−2^
miR-212-3p	*FAM76B*	−0.71	1.60 × 10^−2^
miR-224-5p	*SLC25A36*	−0.71	1.60 × 10^−2^
miR-142-3p	*VPS13A*	−0.70	1.90 × 10^−2^
miR-142-5p	*VPS13A*	−0.68	2.30 × 10^−2^
miR-142-3p	*SIAH1*	−0.65	3.30 × 10^−2^
miR-142-5p	*CCNT2*	−0.63	4.10 × 10^−2^

**Table 12 genes-11-01143-t012:** Statistically significantly negatively correlated miRNA–mRNA pairs for the comparison of asthmatic horses in remission and control horses. Pearson’s correlation coefficient and adjusted *p*-values are indicated.

DE miRNA	DE mRNA	Pearson’s Correlation Coefficient	P_adjusted_
eca-miR-142-3p	*ADAM22*	−0.92	1.00 × 10^−3^
eca-miR-142-3p	*GORAB*	−0.78	1.70 × 10^−2^
eca-miR-142-5p	*ARID1B*	−0.71	3.30 × 10^−2^
eca-miR-142-3p	*NR3C2*	−0.69	3.70 × 10^−2^

**Table 13 genes-11-01143-t013:** Significant negatively correlated miRNA–mRNA pairs for the comparison of asthmatic horses in an exacerbated state and asthmatic horses in remission. Pearson’s correlation coefficient and adjusted *p*-values are indicated.

DE miRNA	DE mRNA	Pearson′s Correlation Coefficient	P_adjusted_
eca-miR-146b-5p	OSGIN2	−0.82	4.7 × 10^−2^
eca-miR-146b-5p	UBTD2	−0.82	4.7 × 10^−2^

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
