# Peer review of "An Integrative miRNA-mRNA Expression Analysis Reveals Striking Transcriptomic Similarities between Severe Equine Asthma and Specific Asthma Endotypes in Humans"

_genes, 2020, doi:10.3390/genes11101143_

Round 1
Reviewer 1 Report
I would like to congratulate the authors for their meticulous work. The methodology, the results and the writing of the article is very clear and concise.
I have a couple of minor comments:
In the figure A8 (PCA of the RNASeq samples), It may be a sample in remission that could be an outlier. This has led me to look at figure A1 (PCA of the miRNASeq) to see that there is also a strange sample in remission (along with another control sample + exacerbation). These samples in remission are from the same horse?.
I think that adding labels (sample name) to each point of the PCA would help.
By the way. The 3 samples shown together in figure A1 belong to the same batch ? In the ms it's explained that the batch is corrected for this (in the statistics I guess), but if the effect is this big, it could be a good idea to use combat and redo the PCA and check the effect.
There is little consistency in the figures, for example A4 and A2 have the names of the miRNAs in the MA plot and the volcano, but in the rest of the figures (A3, A9, A11 ...), the names of the genes do not appear. I think that putting these names to everything (including PCA), would help enormously.
Author Response
Point 1: I would like to congratulate the authors for their meticulous work. The methodology, the results and the writing of the article is very clear and concise.
Response 1: We would like to thank the reviewer for careful and thorough reading of this manuscript and for the thoughtful comments and constructive suggestions, which help to improve the quality of this manuscript. Our response follows:
Point 2: In the figure A8 (PCA of the RNASeq samples), It may be a sample in remission that could be an outlier. This has led me to look at figure A1 (PCA of the miRNASeq) to see that there is also a strange sample in remission (along with another control sample + exacerbation). These samples in remission are from the same horse?.
I think that adding labels (sample name) to each point of the PCA would help.
Response 2:
We would like to thank the reviewer for pointing this and we have made necessary changes to the figures.
The remission sample mentioned in the mRNA PCA is horse ID: 312. Previously we included all genes in the PCA (Figure A8). We determined that sample 312 was ‘different’ from other remission samples due to low variation genes that were included in the PCA. However, as recommended in the DESeq2 manual, we made a PCA of the top 500 most variable genes. This shows that individual 312 clusters tightly with other remission samples indicating that sample 312 is not an outlier, and was only different from other remission samples (on PC2) due to low variation genes.
In addition, to measure the influence of each individual on all genes, we calculated cooks distances for each sample for all genes and represent them as boxplots. We found little to no discernable difference when comparing the distribution of cooks distances for sample 312 to other samples in our dataset. This suggests that sample 312 is not more influential on the individual gene linear models then other samples.
The cooks distances indicate that sample 312 is not a consistent influential outlier across the 20288 genes analyzed, and we do not consider sample 312 an outlier as it clusters tightly with other remission samples in the PCA of the 500 most variable genes.
The remission sample mentioned in the miRNA PCA plot is horse ID: 361 (not the same as 312).
Changes:
We replaced Figure A8 with the PCA of the top 500 most variable genes and label individuals.
We replaced Figure A1 with a PCA labeling individuals.
We included an additional Figure (A9) to show cooks distances for each sample across all genes.
We updated the code on Github to include all the code used to generate the new/adjusted figures
Point 3: By the way. The 3 samples shown together in figure A1 belong to the same batch ? In the ms it's explained that the batch is corrected for this (in the statistics I guess), but if the effect is this big, it could be a good idea to use combat and redo the PCA and check the effect.
Response 3: The three samples mentioned are horse IDs: 361, 001, and 355. These are not part of the same batch, so we know that the processing batch does not explain the clustering of these three samples. Individual 361 is from a separate processing batch from 001 and 355. We colored samples in the PCAs according to their extraction and processing batch for mRNA and miRNA. Sample 001 and 355 are part of batch group 3. Group 3 is evenly distributed across the miRNA PCA plot.
Point 4: There is little consistency in the figures, for example A4 and A2 have the names of the miRNAs in the MA plot and the volcano, but in the rest of the figures (A3, A9, A11 ...), the names of the genes do not appear. I think that putting these names to everything (including PCA), would help enormously.
Response 4: We thank the reviewer for the constructive suggestion regarding consistency in the mentioned plots. We agree and therefore adjusted the corresponding plots to increase consistency.
Changes:
We have added Labels to MA-Plots and Volcano Plots in A3, A9, A10, and A11.
We have increased consistency in Plots A3,A4,A5,A9,A10, and A11 with changing the Plot Title.
As mentioned above, we have also added labels to the miRNA and mRNA PCA Plots (Figure A1 and Figure A8)
Reviewer 2 Report
The manuscript presents an interesting study evaluating how the expression mRNA and miRNA can vary in relation to Severe equine asthma (sEA) using high-throughput data. Moreover, both datasets are integrated to identify target miRNAs and giving a boarder view of how miRNAs-mRNAs interactions are implicated in sEA exacerbate state and remission. The article is scientifically soundness and all the methodology is clearly stated.
I have a few minor suggestions.
First, I would like the authors to further explain the results for the functional enrichment analyses, the title of this section is "Pathway analysis and functional enrichment of differentially expressed mRNAs", however, they describe a Gene Ontology enrichment analysis. If a pathway analysis was performed, the should indicate the database used and the results.
Moreover, I would like to see a further explanation with the number of GO terms enriched in each comparison, the highest enriched terms, with the p-value, number of genes. In addition, it seems that some of the differentially expressed miRNAs are differentially expressed due to the high variability within each condition (eg. eca-miR-31 or eca-miR-223 in the contrasts exacerbation vs. controls). I would like the authors to comment on this in the discussion, is this high variability is expected ( in this case in exacerbation), if the amount of samples used in the analysis is enough to conclude that there are changes in expression for these specific miRNAs among both conditions, what can be done to confirm these results.
Some of the genes in the supplementary material are not in italics.
Author Response
Point 1: The manuscript presents an interesting study evaluating how the expression mRNA and miRNA can vary in relation to Severe equine asthma (sEA) using high-throughput data. Moreover, both datasets are integrated to identify target miRNAs and giving a boarder view of how miRNAs-mRNAs interactions are implicated in sEA exacerbate state and remission. The article is scientifically soundness and all the methodology is clearly stated.
Response 1:
We appreciate the time and effort that you dedicated to providing feedback on our manuscript and are grateful for the insightful comments to improve our paper.
Point 2: First, I would like the authors to further explain the results for the functional enrichment analyses, the title of this section is "Pathway analysis and functional enrichment of differentially expressed mRNAs", however, they describe a Gene Ontology enrichment analysis. If a pathway analysis was performed, the should indicate the database used and the results.
Response 2:
We would like to thank the reviewer for the very attentive input.
We have carried out pathway analysis for the miRNA differential expression analyses results with the tool mirPath v3. On the other hand, we carried out GO-term enrichment analysis for the mRNA differential expression analysis results. To better reflect the analyses we have performed, we adjusted the corresponding subheading 3.7 at Line 338.
Changes:
[Line 338]: We have adjusted the subheading to better reflect the analysis we have performed. The new subheading reads “Gene ontology (GO) term enrichment analysis of differentially expressed mRNAs”
Point 3: Moreover, I would like to see a further explanation with the number of GO terms enriched in each comparison, the highest enriched terms, with the p-value, number of genes.
Response 3:
To have the results of our GO-term enrichment analysis more accessible to the reader, we agree with the reviewer and moved the presentation of the results with the corresponding figure from Supplementary File 2 (Supplementary Note S6) to the main text. Therefore, we do have a more in-depth presentation of the number of enriched GO-terms in the main text together with a figure (new Figure 4) detailing the number of enriched GO terms in each comparison and their p-values.
Changes:
[Line 340] We have expanded the presentation of the GO term enrichment analysis by moving the text from Supplementary note S6 to Line 340.
[line 351] We added new Figure 4 (previously Supplementary Figure S5) in the main text.
Therefore we had to adjust the numbering and in-text references of the following existing figures:
[line 333] Small visual correction in Figure 3
[line 365 & 384] Changed numbering of Figure 4 to Figure 5 and adjusted the in-text reference.
[line 393 & 398] Changed numbering of Figure 5 to Figure 6 and adjusted the in-text reference.
[line 408 & 413] Changed numbering of Figure 6 to Figure 7 and adjusted the in-text reference
Point 4: In addition, it seems that some of the differentially expressed miRNAs are differentially expressed due to the high variability within each condition (eg. eca-miR-31 or eca-miR-223 in the contrasts exacerbation vs. controls). I would like the authors to comment on this in the discussion, is this high variability is expected ( in this case in exacerbation), if the amount of samples used in the analysis is enough to conclude that there are changes in expression for these specific miRNAs among both conditions, what can be done to confirm these results.
Response 4:
We agree with the reviewer that this observation is well worth discussing in the main text of the manuscript.
As we have mentioned in the introduction, severe equine asthma is a highly heterogenous disease, similar to human asthma. Heterogenous diseases can have high within group variability in gene expression.
The signal of the miRNAs mentioned is not driven by one outlier, rather multiple individuals contribute to the within group variation. Thus, we consider our results to be reasonable and think that our results provide a good starting point for future research in sEA pathophysiology.
Validation of results:
Further validation of our results in a new study cohort could be beneficial. RT-qPCR could be applied in the future to validate our findings (both technically & statistically) on our differentially expressed miRNAs and genes.
Sample size:
Generally, differential gene expression analyses with 6 or more samples per group are considered to be robust approaches. However, an increased sample size potentially brings several advantages such as increased statistical power and therefore the ability to detect more differentially expressed genes and miRNAs [3].
Changes:
[line:518] We added a part in the discussion to talk about the within group variability observed
[line: 596] Added section at the end of the discussion as a potential ‘outlook’. There we discuss sample size and potential future experiments for validation and novel experiments.
[line: 951] We have additionally updated the bibliography to include the new reference.
References:
- Papi, A.; Brightling, C.; Pedersen, S. E.; Reddel, H. K. Seminar Asthma. 2018, 783–800, doi:10.1016/S0140-6736(17)33311-1.
- Couetil, L.; Cardwell, J. M.; Leguillette, R.; Mazan, M.; Richard, E.; Bienzle, D.; Bullone, M.; Gerber, V.; Ivester, K.; Lavoie, J. P.; Martin, J.; Moran, G.; Niedźwiedź, A.; Pusterla, N.; Swiderski, C. Equine Asthma: Current Understanding and Future Directions. Front. Vet. Sci. 2020, 7, 1–21, doi:10.3389/fvets.2020.00450.
- Schurch, N. J.; Schofield, P.; GierliÅ„ski, M.; Cole, C.; Sherstnev, A.; Singh, V.; Wrobel, N.; Gharbi, K.; Simpson, G. G.; Owen-Hughes, T.; Blaxter, M.; Barton, G. J. How many biological replicates are needed in an RNA-seq experiment and which differential expression tool should you use? Rna 2016, 22, 839–851, doi:10.1261/rna.053959.115.
Point 5: Some of the genes in the supplementary material are not in italics.
Response 5:
We carefully went through the supplementary material and put all occurrences of genes in italics.
Changes:
[Supplementary File 3, Table S5] We put all genes in italics
[Supplementary File 3, Table S6] We put all genes in italics
[Supplementary File 3, Table S7] We put all genes in italics
[Supplementary File 3, Table S8] We put all genes in italics
[Supplementary File 3, Table S9] We put all genes in italics
[Supplementary File 2, Supplementary Note S3] We put all genes in italics
[Supplementary File 2, Supplementary note S5] We put all genes in italics
